# Structural Modifications of Sodium Polyacrylate-Polyacrylamide to Enhance Its Water Absorption Rate

Ting Xu , Wenxiang Zhu and Jian Sun *

College of Environmental and Chemical Engineering, Shanghai University, Shanghai 201900, China
* Correspondence: sunjkl@shu.edu.cn

**Abstract:** Superabsorbent polymers (SAPs) can absorb a large amount of water and find broad applications in various industries. There are many reports on the synthesis and structural modification techniques to improve the water absorption property of SAPs. However, we see few studies on the comparison and integration of these techniques. In this study, three structurally modified SAPs were synthesized and are evaluated for the effects of surface cross-linking, foaming, and the integration of the two modifications to improve the absorption rate and capacity of a conventional SAP. First, sodium polyacrylate-polyacrylamide was prepared as the base polymer by the aqueous solution polymerization method. Second, the base polymer was modified with surface cross-linking to enhance absorption, and a surface cross-linked SAP was obtained. Third, the base polymer was modified with foaming to obtain a foamed SAP. Lastly, the foamed SAP was modified further with surface cross-linking to obtain a foamed and surface cross-linked SAP. In comparison with the base polymer, the three synthesized SAPs were evaluated for their absorption performance. Results show that their absorption performance could be improved by the three modification processes. Specifically, the surface cross-linked SAP had the best absorption capacity under load, the foamed SAP had the highest absorption capacity of 1954 g/g, and the foamed and surface cross-linked SAP had the fastest absorption rate with an initial swelling rate of $K_{is} = 21.94$.

**Keywords:** acrylamide; surface cross-linking; foaming; water absorption capacity; water absorption rate



## 1. Introduction

Superabsorbent polymers (SAPs) are a new functional material that can absorb a large amount of water. The weight of a typical water-saturated SAP can reach several hundred times its original weight, and it has excellent water retention even under pressure [1,2]. Due to its superior water absorption properties, SAP has broad applications in agriculture, pharmaceuticals, and the medical and health industries [3–5]. The most popular products of SAP include baby diapers and other hygiene products, such as adult diapers, sanitary napkins, and absorbent pads used in hospitals [6,7]. There are numerous recent research studies on sodium polyacrylate SAPs [8,9]. SAPs must have high water absorption capacity and strong salt resistance, and fast water absorption rates in some applications. While many studies were devoted to improving the water absorption capacity of SAPs, the goal of this study was to improve their water absorption rate for applications in diapers through the better understanding of the mechanism and integration of two popular SAP structural modification techniques.

The surface cross-linking technique is desirable to improve the water absorption performance of SAPs, and the method was first proposed by Fredric [10]. SAP particles could be prepared in a core-shell structure to improve the strength and swelling rate of expanded hydrogel by surface cross-linking with a cross-linking agent mixture [11,12]. In water absorption, the absorption rate of traditional SAPs is slower because of the gel blockage

problem after absorption. On the other hand, carboxyls on the SAP surface can be cross-linked by surface cross-linking to form a highly cross-linked shell that allows for a SAP to maintain its shape during expansion [13], and reduces the gel blockage problem of traditional polymers, thus enhancing absorption performance. Therefore, the water absorption rate of the traditional SAP can be enhanced by surface cross-linking modification.

A large number of studies reported on improving the water absorption of SAPs with a surface cross-linking technique. Azizi et al. [14] adopted polyamine modifiers to treat the surface of terpolymers in the presence or absence of an $AlCl_3$ catalyst. Their results showed that the absorbency under load of the terpolymer was effectively improved by 25%. Chang et al. [15] prepared SAPs by cross-linking the surface region of SAP spheres with ethylene glycol diglycidyl ether (EGDE) to improve their mechanical properties. Lee et al. [13] used polycations instead of chemical cross-linking to obtain physically surface cross-linked polyacrylic-acid-based SAP microspheres within 20 min. Kwon et al. [16] synthesized an SAP by copolymerizing itaconic acid and vinyl sulfonic acid, and the absorption performance was improved by introducing surface cross-linking.

Increasing the surface contact between SAP and water is effective in accelerating water absorption by the polymer. The specific surface area could be enhanced by the generation of a porous structure in SAP to absorb a large amount of water in a short amount of time [17]. There are various methods to prepare the porous structure of SAPs, such as phase inversion [18], freeze-drying and hydration, water-soluble porogens, and foaming [19]. Kabiri et al. [20] utilized acetone and sodium bicarbonate as the porogens to prepare a high-porosity structure by optimizing reaction conditions. Chen et al. [21] synthesized porous poly sodium acrylate-coacrylamide with methanol, propanol, and butanol as the foaming agents. Bao et al. used various types of a surfactant and foam stabilizer system, and examined their influence on the porous polyacrylate sodium superabsorbent resin [22]. Researchers also tried to innovate on the foaming system. Si et al. successfully prepared a porous sodium polyacrylate with a new foaming system [23]. Qing et al. [24] studied the effect of two foaming techniques (gas blowing and porosigen) on the physical and swelling properties of the polymers. A drying technique was also studied. Barajas-Ledesma et al. [25] used five drying techniques to prepare porous SAPs. Cao et al. [26] used water vapor released from the dehydration of $Al(OH)_3$ to prepare a porous acrylamide copolymer.

Because of its large surface area, tunable pore structure, high swelling rate, and chemical diversity, porous SAP has received much research attention [27]. However, there are still issues in preparing porous SAPs with foaming. For example, the product purity is low due to the use of additives, the porous structure is uneven, and the pores easily collapse. Adding surfactants and foam stabilizers is an effective way to improve the process. $NaHCO_3$ is widely used as the foaming agent. Sodium dodecylbenzene sulfonate (SDBS) was selected as the surfactant because SDBS foam films manifest high stiffness and low viscoelasticity [28]. Pluronic F127 (PF127) is a common foam stabilizer that can improve the foam stability of the reaction system. Therefore, $NaHCO_3$, SDBS, and PF127 were selected as the foaming system in this study.

This study investigates the effects of three structural modification approaches, surface cross-linking, foaming, and foaming and surface cross-linking, on the absorption performance of SAPs. Four sodium polyacrylate-polyacrylamide (PAANa-PAM) SAPs were prepared using the aqueous solution polymerization method with the reported monomer, initiator, and cross-linking agent [29], as follows: conventional SAP0, surface cross-linked SAP1, foamed SAP2, and foamed and surface cross-linked SAP3. There are numerous reports on the synthesis and structural modifications of SAPs, including surface cross-linking and foaming. However, few studies are on the comparison and integration of these techniques. There is a need for such investigations, and this study could initiate the process.

## 2. Experimental

### 2.1. Materials

Acrylic acid (AA, >99.0%), acrylamide (AM, 99.0%), aluminum trichloride (AlCl$_3$, 99.0%), sodium hydroxide (NaOH, 96.0%), and sodium chloride (NaCl, 99.8%) were supplied by Aladdin, China; ammonium persulfate (APS, 98.5%) and N, N′-methylene-bis-acrylamide (MBA) were purchased from Macklin, China; stabilizer Pluronic F127 (PF127, 98.0%) was from Sigma, USA; Sodium bicarbonate (SHC, 98.0%) and methanol (CH$_3$OH, 99.9%) were obtained from the Sinopharm Group, China; surfactant sodium dodecylbenzene sulfonate (SDBS, 90.0%) was purchased from Zhiyuan, China; ethylene glycol diglycidyl ether (EGDE, 99%+) was from Adamas, China. All materials were analytical-grade.

### 2.2. Sample Preparations

Sample preparation processes are shown in Figure 1. Sodium bicarbonate, sodium dodecylbenzene sulfonate, and Pluronic F127 served as the foaming agent, surfactant, and foam stabilizer, respectively. The surface cross-linking liquid consisted of AlCl$_3$, ethylene glycol diglycidyl ether, CH$_3$OH, and deionized water. The specific preparation processes of SAP0, SAP1, SAP2, and SAP3 were as follows:

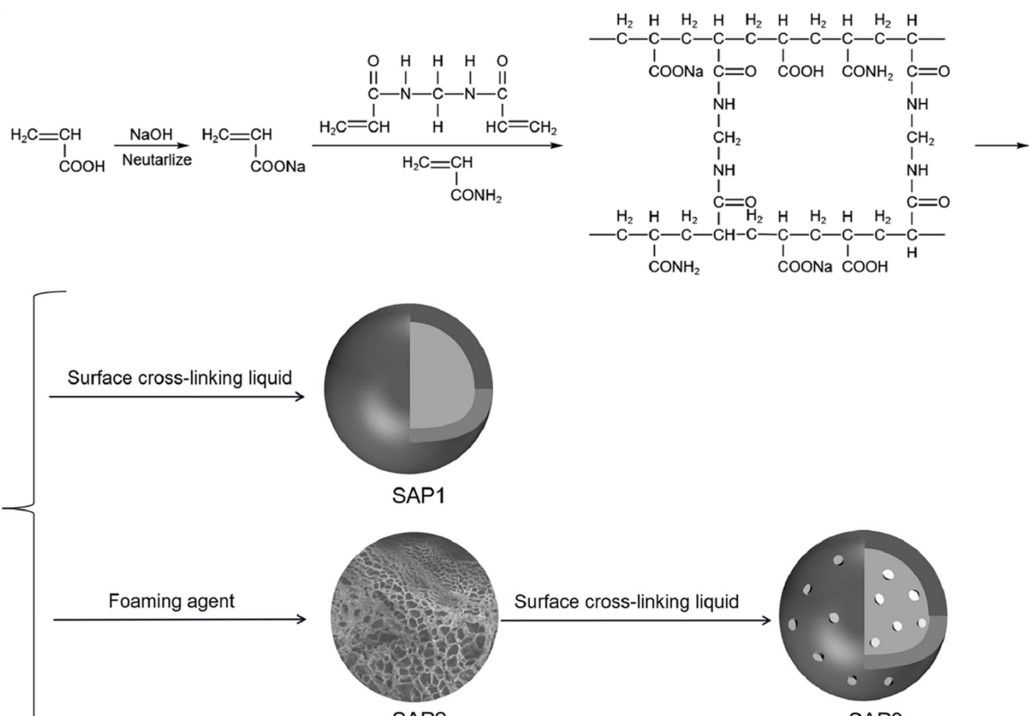

**Figure 1.** Preparation processes of SAP0, SAP1, SAP2, and SAP3.

The conventional PAANa-PAM (SAP0) was prepared as the base absorbent in this study following the literature [29]. First, a 5.7 mL 25% NaOH solution was slowly added to 5 g undiluted acrylic acid under stirring in a three-necked flask equipped with a thermometer. The mixture was stirred at 20 °C for 30 min to obtain an acrylic acid solution with a neutralization degree of 65%. Second, 22.5 mL deionized water, 2.5 g acrylamide, and 0.0008 g N, N′-methylene-bis-acrylamide were added into the mixture and stirred in the nitrogen atmosphere for 30 min. Third, the system was heated to 70 °C, 0.0175 g ammonium persulfate was added, and the reaction was kept at 70 °C for 2 h. After reaction and cooling, the resulting gel was removed from the reaction vessel, cut into small patches, and dried in an oven at 50 °C to a constant weight. Lastly, the dry product was crushed with a pulverizer (Changzhou Yuexin Instrument Manufacturing Co., Ltd., Changzhou, China) and milled through a mesh screen to obtain SAP0.

SAP0 was modified with the surface cross-linking process as reported [30] to obtain the surface cross-linked PAANa-PAM (SAP1). The crushed SAP0 was sprayed with 1.21 g surface cross-linking liquid while stirring. Subsequently, it was placed in a vacuum drying oven to react at 120 °C for 3 h and then sifted with a 40 to 120 mesh sample sieve. The usage amounts of $AlCl_3$, ethylene glycol diglycidyl ether, and $CH_3OH$ were optimized and were 10, 2, and 9 wt.%, respectively.

SAP0 was modified with a foaming process as reported [31] to obtain the foamed PAANa-PAM (SAP2). In the synthesis of SAP0, 0.0523 g of a surfactant (sodium dodecylbenzene sulfonate) and 0.0125 g of a foam stabilizer (Pluronic F127) were added to the mixture while adding deionized water, acrylamide, and N, N′-methylene-bis-acrylamide separately. After adding ammonium persulfate, 0.09 g of a foaming agent (sodium bicarbonate) was poured into the system. The other steps followed the process of SAP0.

SAP2 was modified with surface cross-linking to obtain the foamed and surface cross-linked PAANa-PAM (SAP3) with the same procedure as that for SAP1.

### 2.3. Water Absorption Capacity, Absorption Rate, and Retention Rate

The measurement of water absorption capacity in this study followed a procedure reported in the literature [32]. An approximate 0.5 g of the dried SAP sample was immersed and allowed to swell in deionized water or a NaCl solution (0.9 wt.%). The swollen sample was then taken out, and nonabsorbed water was removed with a 100-mesh gauze. Sample weight was measured, and absorption capacity was calculated with the following equation:

$$Q(g/g) = \frac{M_2 - M_1}{M_1} \tag{1}$$

where $M_1$ and $M_2$ are the weights (g) of the dry and saturated samples, respectively. The absorption capacity of deionized water/NaCl solution is represented by $Q_w$/$Q_s$, respectively.

Following the method above, the water absorption capacity of the samples was measured every two minutes until the absorption equilibrium had been reached [33]. Water absorption rate over time was then obtained with time as the abscissa and absorption capacity as the ordinate.

For practical application, the absorption under load (*AUL*) and centrifuge retention capacity (*CRC*) of SAP samples were also measured. The sample was weighed ($W_0$, g) and uniformly dispersed on the surface of polyester gauze on a macroporous sintered glass filter plate in a Petri dish. A cylindrical solid load (0.3 psi) was placed on the sample, and 0.9 wt.% of a NaCl solution was added onto the Petri dish. After 1 h, the nonabsorbed solution was removed, and the swollen sample was weighed ($W_1$, g). The *AUL* (g/g) value was calculated as follows [34]:

$$AUL = \frac{W_1 - W_0}{W_0} \tag{2}$$

The sample ($\omega_0$, g) was weighed and soaked in a 0.9 wt.% NaCl solution for 30 min. It was then removed from the residual solution and centrifuged at 300 G. The swollen sample ($\omega_1$, g) was weighed, and the *CRC* (g/g) was calculated as follows [25]:

$$CRC = \frac{\omega_1 - \omega_0}{\omega_0} \tag{3}$$

### 2.4. SEM, FTIR, and TG Analyses

The morphology of the dried gel structure of SAP samples was studied with SEM [35]. Samples were freeze-dried, coated with a thin layer of palladium–gold alloy, and imaged with a SEM (HITACHI, Tokyo, Japan). The FTIR characterization of the SAP samples was performed on a Nicolet-is50 FTIR (Thermo Fisher Scientific, Waltham, MA, USA) using potassium bromide tablets. The thermal stability of the SAP samples was studied on an

STA449F3 thermal analyzer (Netzsch-Geratebau GmbH, Selb, Germany) in a nitrogen atmosphere. Samples were heated from 30 to 800 °C at a heating rate of 20 °C/min.

## 3. Results and Discussion

### 3.1. FTIR Analyses

The full and enlarged FTIR spectra of the SAP samples are shown in Figure 2. In Figure 2a, the broad and strong peaks around 3451 cm$^{-1}$ are attributed to the stretching vibration of the N–H of the amide group and O–H of the carboxylic acid group [36,37]. Those at 2938~2918 cm$^{-1}$ could be attributed to the C–H stretching vibration [38]. The absorption peaks at 1671~1638 cm$^{-1}$ were assigned to the C=O stretching vibration [39] in the amide group, which demonstrated that AM monomers were all successfully introduced into the SAP structure. In Figure 2b, the peaks around 1580~1570 and 1413~1400 cm$^{-1}$ could be assigned to the C=O asymmetric and symmetric stretching vibration, which also corresponded to the characteristic –COO absorption [16,40] in carboxylates. This indicated that the partially neutralized AA was successfully introduced into the polymer. In addition, SAP1, SAP2, and SAP3 had much stronger absorption peaks than those of SAP0 in 1580~1570 cm$^{-1}$ because the polymerization degree of PAANa-PAM SAPs was increased by structural modification.

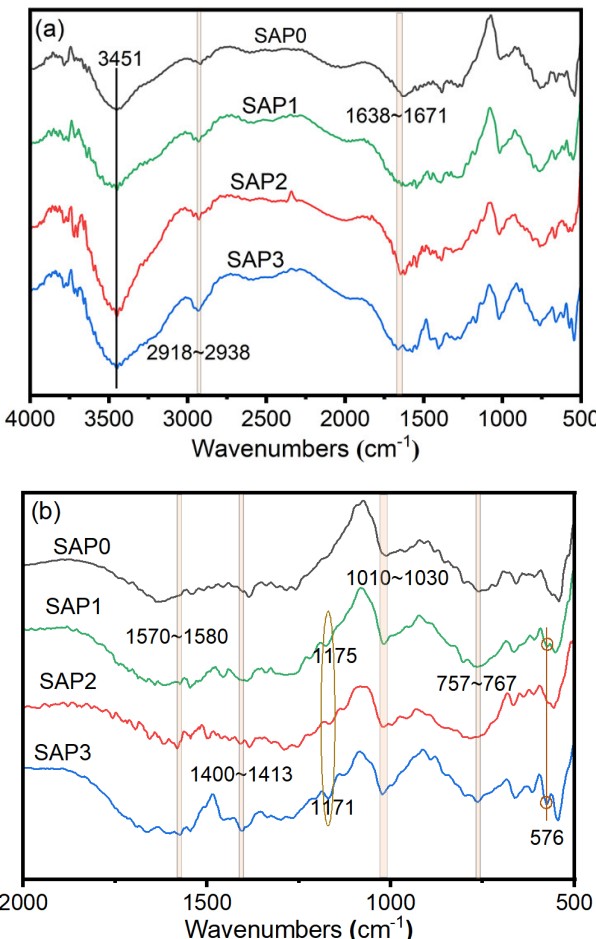

**Figure 2.** (a) Infrared spectra from 4000 to 500 cm$^{-1}$ and (b) enlarged infrared spectra from 2000 to 500 cm$^{-1}$ of SAP0, SAP1, SAP2, and SAP3.

A new peak appeared at 1170 cm$^{-1}$ in SAP1 and SAP3, which was attributed to the C–O–C asymmetric stretching vibration generated by the ring opening addition of the oxirane rings of EGDE and the –COOH group [41], indicating the progress of the surface cross-linking reaction. The absorption peaks around 1020 cm$^{-1}$ belonged to the C–O

stretching vibration [42], and the band at 1015 cm$^{-1}$ in SAP0 shifted to higher wavenumbers 1022 cm$^{-1}$ in SAP1 and SAP3, which also indicated the formation of a new C–O–C bond after surface cross-linking. Those at 767~757 cm$^{-1}$ were ascribed to –CH$_2$ and –CH. The peaks around 576 cm$^{-1}$ in SAP1 and SAP3 could be assigned to Al–O–C due to the formation of coordination complexes between Al$^{3+}$ and –COO$^-$ in the polymer after surface cross-linking [43]. The spectral analyses above confirmed the successful synthesis of four PAANa-PAM SAPs.

### 3.2. SEM Analyses

All SAP samples were characterized by SEM, and the photos of their surface morphology are shown in Figure 3. The intact SAP0 particle presented a typical morphology, sometimes known as riverlike, for the brittle plastics. After surface treatment, a slightly changed morphology with a rougher surface and some small fissures on the surface of SAP1 were obtained. In surface cross-linking, the cross-linking agent reacted with the surface groups of the polymer, increasing the cross-linking density and forming a "core-shell" structure, which was beneficial to alleviating gel blockage. SAP2 exhibited a smooth, compact, and uniformly sized and interconnected porous structure formed by combined actions of the foaming agent, surfactant, and foam stabilizer. The porous structure boosted the surface area of SAP for retaining more water. SAP3 also showed a rough surface and a porous structure. However, compared with SAP2, which maintained better porosity, SAP3 lost the uniformity of a porous structure. Both SAP1 and SAP3 showed rough surfaces, while the surface of SAP1 was compact without pores. SAP3 had a large number of pores, indicating that the specific surface area of SAP3 should be larger than that of SAP1. The results above confirm the successful synthesis of four PAANa-PAM SAPs.

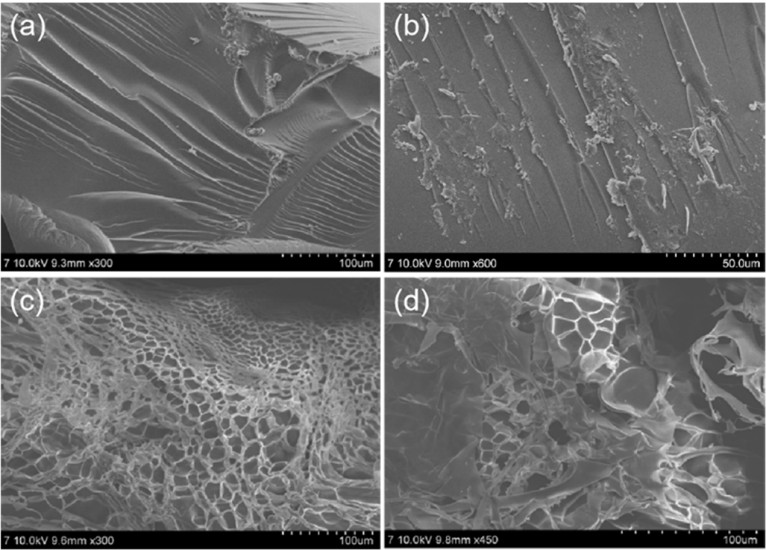

**Figure 3.** SEM images of (**a**) SAP0, (**b**) SAP1, (**c**) SAP2, and (**d**) SAP3.

### 3.3. TG Analyses

Figure 4 shows the TG curves of the four SAPs. Their thermal decomposition could be roughly divided into five stages. The first stage was around 30–185 °C; mass loss was due to the loss of free water and bound water. The second stage was around 186–296 °C, and mass loss was due to the additional loss of moisture. The third stage occurred around 297 °C, and ended at 387, 393, 380, and 401 °C for SAP0, SAP1, SAP2, and SAP3, respectively, attributed to the decomposition and oxidation of the carbohydrate chain [44]. The fourth stage started at 388, 394, 381, and 402 °C and completed at 603, 569, 617, and 591 °C for SAP0, SAP1, SAP2, and SAP3, respectively. The mass loss of this stage was caused by the decomposition of carboxyl and amide groups. The last stage occurred at 604, 570, 618, and 592 °C and all

ended at 784 °C. The mass loss could be attributed to the fracture and decomposition of the polymer backbone [45].

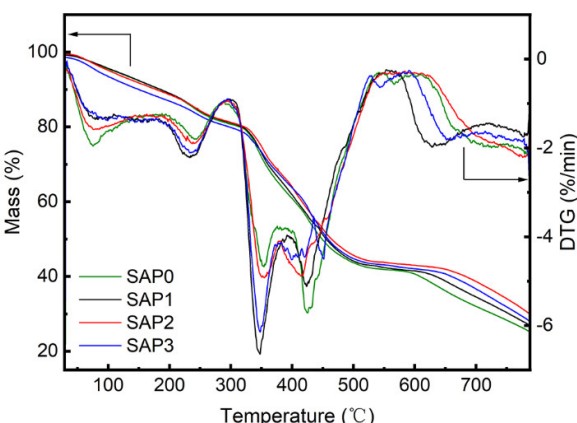

**Figure 4.** Thermogravimetric analysis curves of SAP0, SAP1, SAP2, and SAP3.

Mass losses of the three modified SAPs were slower than that of SAP0, as shown in Figure 4. The temperature for the mass retention rate to be reduced by 50% was 448, 455, 468 and 459 °C for SAP0, SAP1, SAP2 and SAP3, respectively. At 784 °C, the mass retention rates of SAP0, SAP1, SAP2, and SAP3 were 25.5%, 27.0%, 30.8%, and 28.0%, respectively. Therefore, SAP1, SAP2 and SAP3 showed better thermal stability than that of SAP0 at high temperatures, and SAP2 had the best thermal stability. SAP1 showed higher thermal stability than that of SAP0 due to the increase in cross-linking degree and the formation of a denser network, reducing the chain mobility of SAP and resulting in a less flexible network [16,46]. The thermal stability of SAP2 was also higher than that of SAP0. During the process of the cross-linking polymerization of SAP2, the dispersion and dissolution of $NaHCO_3$ increased the pH of the reaction medium [17], thus increasing the initiator decomposition extent [47] and promoting the polymerization reaction rate. Therefore, the molecular weight of the product and the chain length inside the polymer were increased to improve the thermal stability. In addition, the thermal stability of SAP3 was lower than that of SAP2, presumably due to the excessive cross-linking of SAP3. In general, all three modified SAPs showed good thermal stability, with SAP2 being the best.

*3.4. Absorption Capacities*

The absorption capacities of unsalted and salted water ($Q_w$ and $Q_s$) for the SAP samples were measured and are shown in Figure 5. Each sample was tested three times and calculated for the average with a small variance. The absorption capacity of SAP0 was greatly improved after modifications. $Q_w$ was increased for SAP1 by surface cross-linking with SAP0. Conventional polymer particles were easy to bond into agglomerates after coming into contact with water, resulting in gel blockage and impeding the further penetration of water molecules. This phenomenon was minimized, and water absorption capacity was increased after surface cross-linking. The highest $Q_w$ was achieved with the foaming modification for SAP2 because of the synergistic effect of surfactant SDBS and foam stabilizer PF127, which produced low packing density and high foam stability, resulting in a better foaming effect of $NaHCO_3$ and the uniform formation of a porous structure [22]. The porous SAP2 produced more space to absorb more water to a certain extent [48], so the absorption capacity of the gel was effectively enhanced. The $Q_w$ of SAP3 prepared with foaming and surface cross-linking was lower than that of SAP2, presumably because the absorption expansion of SAP3 was restricted by the high degree of surface cross-linking. In addition, the salt resistance of SAP0 was significantly improved by all three modification approaches, and SAP2 achieved the best performance.

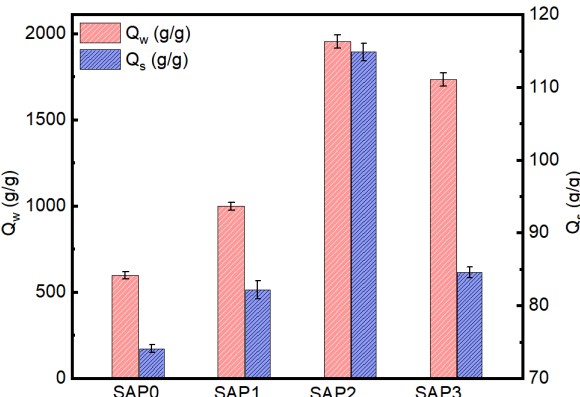

**Figure 5.** Absorption capacities of deionized water and saline water for SAP0, SAP1, SAP2, and SAP3.

SAP1 and SAP3 were more sensitive to salt water absorption. This could be attributed to the following reasons. First, there might have been a shielding effect between the ions in the salt solution and the water absorption functional groups of SAPs that reduced the osmotic pressure that promoted water absorption, thus affecting absorption [49]. Second, the lower $Q_s$ of the surface cross-linked SAP could be attributed to the increase in cross-linking density, which prevented further absorption [13]. Third, SAP3 was obtained by integrating both the modifications of foaming and surface cross-linking to improve the absorption rate. The high tendency to rapidly imbibe water could cause clumping or the formation of a fish eye during absorption [50], preventing water from entering the internal particles, and reducing the absorption capacity. Therefore, SAP3 was the most sensitive to salt water absorption.

$Q_w$ and $Q_s$ for SAP2 prepared in this study reached 1954 and 115 g/g, respectively. In comparison with the reported 1878 and 119 g/g for a typical foamed starch-based macro-porous (St-MP) SAP by Meng [51], the $Q_w$ of SAP2 was slightly higher. This was presumably because SAP2 was obtained by foaming with the foaming agent, surfactant, and foaming stabilizer, while the St-MP SAP synthesized by Meng only adopted a foaming agent. However, the salt resistance of St-MP SAP was slightly higher than that of SAP2, suggesting that the absorption performance of SAPs could be improved by using starch as one of the base materials. In another study for a typical foamed sodium polyacrylate (PAANa) superabsorbent by Bao [22], $Q_w$ was 484 g/g with the use of a single monomer of AA. Bth the basic material and foaming system could greatly impact the properties of foamed SAPs.

*3.5. AUL*

Absorption capacity under load is another important factor in the practical application of SAPs, such as sanitary products. The *AUL* values of SAP samples measured with a 0.9 wt.% NaCl solution are shown in Figure 6. Both SAP1 and SAP3 had higher *AUL* values than those of SAP0, indicating that surface cross-linking was effective in increasing the *AUL* value. SAP1 had the highest *AUL* value and it was strongly related to its network strength, which was improved with its high degree of cross-linking [52]. The *AUL* value of SAP2 was slightly lower than that of SAP0, because the porosity structure reduced the gel strength of SAP2. Hence, surface cross-linking, and foaming and surface cross-linking modifications were beneficial to improving *AUL*, while foaming alone was not.

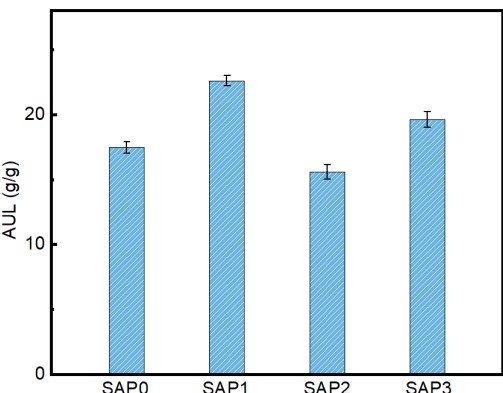

**Figure 6.** *AUL* of SAP0, SAP1, SAP2, and SAP3.

*3.6. Absorption Rates*

The changes in the absorption capacity $Q_w$ of SAP samples over time are displayed in Figure 7a. To better evaluate and quantify their absorption rates, a pseudo-second-order swelling kinetic model [53,54] was used to evaluate the swelling behavior of samples in this study. The model was expressed as follows:

$$\frac{t}{Q_w(t)} = A + Bt \tag{4}$$

where t is the time, and $Q_w(t)$ is the swelling capacity at time t. The $t/Q_w(t)$-t curves of SAP samples were linearly fitted with Equation (4). Fitting results and linear equations are shown in Figure 7b. The theoretical equilibrium swelling capacity ($Q_{w eq,t}$) and initial swelling rate ($K_{is}$) were obtained according to the fitting results in Equations (5) and (6).

$$Q_{w eq,t} = \frac{1}{B} \tag{5}$$

$$K_{is} = K_S Q_{w eq,t}^2 = \frac{1}{A} \tag{6}$$

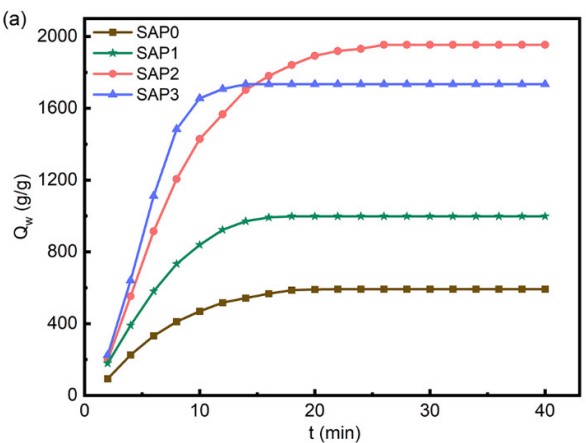
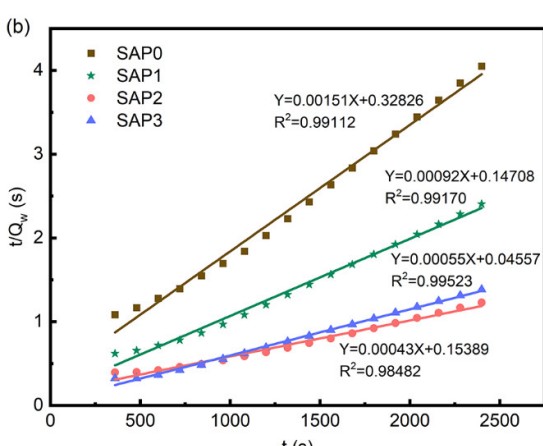

**Figure 7.** (**a**) Absorption rates and (**b**) swelling kinetics of deionized water for SAP0, SAP1, SAP2, and SAP3.

As shown in Table 1, the theoretical values of the equilibrium swelling capacity calculated from modeling were close to those obtained from experiments, indicating that the model achieved good accuracy for all samples. The initial swelling rate ($K_{is}$ = 3.05) of SAP0 was the lowest among all samples.

**Table 1.** Swelling kinetic parameters of SAP0, SAP1, SAP2, and SAP3.

| Sample | $Q_{weq,e}$ [a] | $Q_{weq,t}$ [b] | $K_{is}$ [c] |
|--------|--------|--------|--------|
| SAP0 | 592 | 662 | 3.05 |
| SAP1 | 998 | 1087 | 6.80 |
| SAP2 | 1954 | 2326 | 6.50 |
| SAP3 | 1734 | 1818 | 21.94 |

[a]: $Q_{weq,e}$ is the equilibrium swelling capacity obtained from experiment. [b,c]: $Q_{weq,t}$ and $K_{is}$ were obtained from modeling.

The absorption rate of SAP1 was greater than that of SAP0, as shown in Figure 7a, and the initial swelling rate ($K_{is}$) of SAP1 was greater than that of SAP0 shown in Table 1. This indicated that the absorption rate of the PAANa-PAM SAP was accelerated by surface cross-linking, which formed a "core-shell" structure in the polymer and thus reduced its solubility and increased the water absorption rate. The increase in cross-linking degree was mainly through the formation of coordination bonds between $Al^{3+}$ and $–COO^-$ in the polymer, and the ring opening addition reaction between the epoxy group of EGDE and the –COOH group. The absorption rate of SAP2 was much greater than that of SAP0, as shown in Figure 7a, and its initial swelling rate ($K_{is}$) was greater than that of SAP0 in Table 1. This was because porosity was formed in SAP2, and its specific surface area increased. In addition, surface cross-linked SAP1 and SAP3 reached the absorption equilibrium faster than SAP0 and SAP2 did, as shown in Figure 7a. SAP3 had the highest initial swelling rate ($K_{is}$ = 21.94), as shown in Table 1, which was consistent with the faster absorption rate of SAP3 at the initial stage shown in Figure 7a. This indicates that the absorption rate of SAPs could also be improved by combined foaming and surface cross-linking modifications. In summary, the absorption rate of SAP0 was greatly improved with the three modification techniques. Among all samples, SAP2 had the highest absorption capacity, and SAP3 had the fastest absorption rate.

SAP3 reached its swelling equilibrium in 14 min, in comparison with a foamed and surface cross-linked PAANa prepared by Zhu that reached the maximal absorption capacity in 6 min [55]; SAP3 had higher absorption capacity (1734 g/g) than that of Zhu (1130 g/g). The difference in absorption rate could have been due to the difference in foaming agent and the composition of surface cross-linking liquids. There was a trade-off in achieving high absorption capacity or high absorption rate in the synthesis of SAPs.

*3.7. CRC*

The water retention performance of the SAP samples was studied further with the *CRC* evaluation, as shown in Figure 8. SAP1 had a slightly lower *CRC* value than that of SAP0, indicating that the water retention performance of SAPs would deteriorate after surface cross-linking. This was because increasing the cross-linking density resulted in a decrease in the size of the polymer network. The three-dimensional network structure of SAPs was presumably difficult to stretch during the water absorption process, and the accommodation of free water was reduced. The *CRC* value of the foamed SAP2, and the foamed and surface cross-linked SAP3 was higher than that of SAP0, of which SAP2 was the highest possibly because water was better retained by the moderate polymerization, and more uniform and stable polymer network that was formed in the foamed SAP2. In comparison with SAP1 with surface cross-linking, SAP3 showed better retention performance because of its foaming modification. In comparison with SAP2 with foaming, SAP3 showed worse retention performance because of its surface cross-linking modification. Hence, in this study, foaming modification, and foaming and surface cross-linking techniques improved the water retention performance of SAPs while surface cross-linking alone could not.

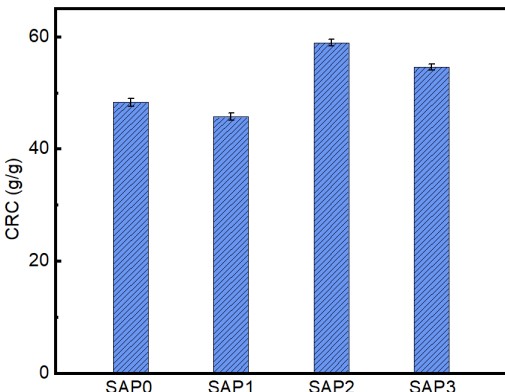

**Figure 8.** *CRC* of SAP0, SAP1, SAP2, and SAP3.

### 3.8. Comparison of Results with the Literature

Table 2 presents a comparison of the absorption performance of SAP1 from this study and those reported in the literature with surface cross-linking modification. SAP1 showed the highest absorption capacity after surface treatment in the saline solution. SAP1 also had a higher *AUL* value than those of Reports 1 and 4, but lower than that of Report 3, which was obtained by improving the cross-linking process on the surface. In comparison with Report 2, SAP1 had a slightly lower *AUL* value but a higher *CRC* value than that of Report 2, probably due to the use of different monomers. Overall, SAP1 showed good water absorption performance, indicating that the surface cross-linking technique was particularly advantageous to increasing the *AUL* value. Comparisons for SAPs 2 and 3 with foaming were not performed due to the lack of data in recent publications.

**Table 2.** Comparison of absorption performance of SAP1 in this study with reported works on surface cross-linking modification.

| Sample | Monomer | $Q_s$ (g/g) | *AUL* (g/g) | *CRC* (g/g) | Reference |
|---|---|---|---|---|---|
| SAP1 | AA, AM | 82.2 | 22.6 | 45.9 | This work |
| Report 1 | AA | 51.0 | 20.7 | / | [34] |
| Report 2 | AA, cellulose, and vinyl sulfonic acid | / | 23.3 | 30.4 | [42] |
| Report 3 | AA | 45.0 | 40.0 | / | [56] |
| Report 4 | AA | 59.0 | 17.6 | / | [57] |

### 4. Conclusions

In this study, four SAP samples were prepared and evaluated for the effects of three structural modification processes on their water absorption and retention performance. Results indicated that surface cross-linking was more effective for increasing the absorption capacity under load, foaming was more beneficial for improving the absorption capacity, while the combination of foaming and surface cross-linking was more advantageous to enhancing the absorption rate. The *AUL* was maximized to the value of 22.6 g/g with the surface cross-linking modification for SAP1. The $Q_w$ and $Q_s$ of the foamed SAP2 were the highest, which were 1954 and 115 g/g, respectively, and its water retention performance was also the best. In addition, the foamed and surface cross-linked SAP3 had the highest absorption rate with an initial swelling rate of $K_{is} = 21.94$ and the second highest absorption capacity. FTIR and SEM studies confirmed the SAP structural modifications, and the three modified SAPs also showed good thermal stability with TG analysis. In summary, the structural modifications of foaming, surface cross-linking, and the integration

of the two proposed in this study could improve the performance of SAP material for commercial applications.

**Author Contributions:** Conceptualization, T.X. and J.S.; methodology, T.X. and W.Z.; software, T.X.; formal analysis, T.X.; data curation, T.X.; writing—original draft preparation, T.X.; writing—review and editing, T.X., W.Z. and J.S.; supervision, J.S.; funding acquisition, J.S. All authors have read and agreed to the published version of the manuscript.

**Funding:** This research was funded by [Shishi Huayong Novel Material Technology Co., Ltd] grant number [D.71-0111-19-046].

**Institutional Review Board Statement:** Not appliable.

**Informed Consent Statement:** Not appliable.

**Data Availability Statement:** Data citation: [dataset]

1.  Bao, J.; Chen, S.; Wu, B.; Ma, M.; Shi, Y.; Wang, X.; Table III. Swelling Kinetic Parameters of M1, M2, M3, M4, and M5; 10.1002/app.41298.
2.  Ghasri, M.; Bouhendi, H.; Kabiri, K.; Zohuriaan-Mehr, M.J.; Karami, Z.; Omidian, H.; Table 3 Comparison of swelling properties of optimized samples in this work (microwave method) with some published works on thermal surface treatment.; 10.1007/s13726-019-00722-6.
3.  Kim, J.S.; Kim, D.H.; Lee, Y.S.; Table 4. The absorption properties of the surface-crosslinked SAP (SSAP) according to the composition of monomer; 10.3390/polym13040663.
4.  Meng, Y.; Ye, L.; Figure 7. Equilibrium swelling ratio as a function of SHC content for St-MP SAP in distilled water (a) and brine (b).; 10.1002/app.44855.
5.  Zhu, S.S.; Xiao, Z.; Zhou, X.D.; Zhao, S.S.; Ye, X.W.; Li, J.Y.; Fig. 5. The rate of different kinds of super absorbent resin absorbing deionized water; 10.3969/j.issn.1001-3539.2019.06.009.
6.  Moini, N.; Kabiri, K.; Zohuriaan-Mehr, M.J.; Omidian, H.; Esmaeili, N.; Table 2. Effects of the epoxy silane (EPS) content and the surface treatment method on free absorbency in distilled water (QDW), free absorbency in saline (QS), saline absorbency under load (AUL) and storage modulus at frequency 1 rad/s (G0) of the swollen samples; 10.1002/pat.4006.
7.  Ghasri, M.; Jahandideh, A.; Kabiri, K.; Bouhendi Hossein Zohuriaan-Mehr, M.J.; Moini, N.; TABLE 2 Surface modification conditions, swelling data, and salt sensitivity factor (f) for surface-crosslinked P (SA-co-AA) with glycerol-lactic acid star-shaped oligomers. The samples were modified at 120 °C for 2 hours, amidosulforic acid used as catalyst (0.2 g); 10.1002/pat.4476.

**Conflicts of Interest:** The authors declare no conflict of interest.

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
