# Peer review of "Structural Modifications of Sodium Polyacrylate-Polyacrylamide to Enhance Its Water Absorption Rate"

_coatings, doi:10.3390/coatings12091234_

Round 1
Reviewer 1 Report
The text entitled "Structural modifications of sodium polyacrylate-polyacrylamide to enhance its water absorption rate", written by Ting Xu, Wenxiang Zhu, Jian Sun, is a study aimed at finding ways to modify superabsorbents in order to improve their properties, in particular capacity and rate of water absorption. This study represents a step forward in comparison with the works listed in the introduction. The main idea put forward by the authors is obvious - with an increase in the contact surface area with water, the absorption rate will grow up. During the sorption process the volume of the granules increases, while the system tends to reduce the free energy and tries to reduce the surface area. If rigid bonds are preliminarily added to the surface, preventing its shrinking, then the absorption rate will remain high, but the capacitance will decrease slightly compared to the case without additional bonds on the surface. This fact was verified experimentally in the work.
In general, the work is worthy of being published after correcting the order of references to the literature (after [47] references comes [51], and then [48], [49] and [50]).
Author Response
Dear Reviewer, Attached please find my responses to your comments. All your guidance and criticism are greatly appreciated. 1. The order of references from [47] to [51] has been corrected. See manuscript page 9 and 10. Thank you. Sincerely yours, Ting Xu Graduate student of Shanghai UniversityReviewer 2 Report
The following report is based on my review of the manuscript entitled “Structural modifications of sodium polyacrylate-polyacrylamide to enhance its water absorption rate”. The manuscript fits within the scope of “coatings” and is also interesting. However, the following shortcomings have been pointed out and need to be addressed properly for further improvement of the manuscript. They are:
1- It is suggested to remove the first sentence in the abstract.
2- Too many abbreviations were observed in the abstract. Kindly reduce/remove all from the abstract
3- To reduce the length of the abstract, it suggested to remove the sentence “Fourier Transform Infrared Spectroscopy (FTIR), Scanning Electron Microscopy (SEM), and thermogravimetric (TG) analyses of the SAPs were performed for better understanding of the structural modification technique”.
4- Some paragraphs are too short in the introduction section. It is suggested to have a minimum of four (4) sentences and a maximum of nine (9) sentences per paragraph.
5- The novelty and practical applicability of this study should be highlighted more in the introduction section.
6- Authors tried to present the structure of the article at the end of the introduction. However, it is too lengthy and not well structured. Kindly improve on that. The novelty of the work alongside the objectives should be highlighted more in the last paragraph of the introduction section. Presently, it contains result. This need to be revised.
7- It is strongly recommended to move Fig. 1 into the methodology section
8- It is suggested to move the sentences in Line 96-100 to the methodology section “The preparation processes were shown in Fig.1. Sodium bicarbonate (SHC), sodium dodecylbenzene sulfonate (SDBS), and Pluronic F127 (PF127) served as the foaming agent, surfactant, and foam stabilizer, respectively. The surface cross-linking liquid consisted of aluminium trichloride (AlCl3), EGDE, methanol (CH3OH), and deionized water”.
9- It is suggested to re-organize the conclusion section much better. The conclusion is too much. Kindly summarise by highlighting the key findings of the study and their implications. Also, there should be no citation in the conclusion section.
10- It is suggested to compare your findings with existing literature on different modifications carried out on sodium polyacrylate-polyacrylamide. A comparison Table is expected between this work and other earlier related published works tabulating the latest works done in this domain to more effectively highlight the novelty of the present work. More explanations and interpretations must be added for the results.
11- Discussion of results is weak. It is suggested to compare the results of the present study with some similar studies. More explanations and interpretations must be added for the results. Authors may refer to these articles to enrich explanations on Figure 2. (a) Infrared spectra 4000 cm-1 to 500 cm-1 and (b) the enlarged of Infrared spectra 2000 cm-1 to 500 cm-1 of SAP0, SAP1, SAP2, and SAP3 and Figure 3. SEM images of (a) SAP0, (b) SAP1, (c) SAP2, and (d) SAP3.
· https://doi.org/10.1007/s13399-022-02431-2
· https://doi.org/10.3390/ijerph18157949
12- The study is rich. However, it is surprising that no optimization was adopted using any of RSM, Uniform Design, Taguchi, or ANN?.
Finally, the article should be modified according to above-said comments and be thoroughly reviewed again before accepting it for publication
Author Response
Dear Reviewer,
Attached please find my responses to your comments. All your guidance and criticism are greatly appreciated.
- It has been removed from Abstract, See manuscript page1, abstract.
- Most abbreviations in Abstract have been removed. See manuscript page1, abstract.
- The sentence has been removed. See manuscript page1, abstract.
- Paragraphs in Introduction have been revised accordingly. See manuscript page1 and 2, introduction, lines 27-97.
- The novelty and practical applicability of this study has been added and highlighted in Introduction. See manuscript page 3, lines 94-97.
- The “results” have been removed and novelty and practical applicability of this study have been added in and highlighted. See manuscript page 2 and 3, lines 88-97.
- 1 has been moved into Sample preparation. See manuscript page 4, line 142.
- This sentence has been moved to Sample preparations. See manuscript page 4, lines 110-114.
- Conclusions have been rewritten and re-organized. Citation has been deleted. See manuscript page 12, lines 385-399.
- A new table has been added. There were only single modification techniques and few comparative studies of different modifications reported in recent existing literature. Majority of the latest researches focused on surface cross-linking, therefore, only the surface cross-linking reports have been compared in the table. See manuscript page 12, lines 371-383.
- More explanations and interpretations have been added to the results of Fig. 2 and Fig. 3. See manuscript page 5 and 6, lines 184-187, 192-194, 203-206, and 209.
- Synthesis optimizations were performed in this study by changing one variable at a time within the ranges reported in the literature. See the sample preparation section, paragraph 3, line 130. However, the optimization processes and data were omitted in discussion to limit the length of manuscript. Trying out some advanced optimization methodology is among our future study.
Thank you.
Sincerely yours,
Ting Xu
Graduate student of Shanghai University
Reviewer 3 Report
The article is very well organized and presented in an efficient way. However, I believe there are some minor errors that should amend before its acceptance.
1. There are numerous English and grammatical mistakes have been found. Please go through the entire manuscript thoroughly.
2. Emphasize novelty both in abstract and introduction.
3. Experimentation should be a little bit clearer. it’s a little confusing. you shouldn’t use abbreviations in this part as you aren’t using standard abbreviations and its confusing which material are you talking about. Like AA, AM etc. I need to go back to materials section every time and see what is this material.
Author Response
Dear Reviewer,
Attached please find my responses to your comments. All your guidance and criticism are greatly appreciated.
- Manuscript has been revised.
- The novelty has been emphasized both in Abstract and Introduction. See manuscript page 1, lines 7 and 94.
- The experimental section has been modified. Contents have been added and abbreviations have been reduced in the Sample preparations section. See manuscript page 3, lines 109-139.
Thank you.
Sincerely yours,
Ting Xu
Graduate student of Shanghai University
Reviewer 4 Report
1- The significance of the work should be highlighted in the last paragraph of the introduction.
2- It is advisable to support sections 2.2&2.3 especially the equations by the suitable references.
3- Please use technical writing in “procedure re- 146 ported in the literature [30]” such as “…based on Author et al (Year)”.
4- How would be the stretching vibration proved in FTIR analyses could facilitate the performance of the polymers.
5- The authors are advised to compare the outcome of the present work with the up to date literature.
Author Response
Dear Reviewer,
Attached please find my responses to your comments. All your guidance and criticism are greatly appreciated.
- It has been highlighted in the last paragraph in Introduction. See manuscript page 2 and 3, lines 94-97.
- Corresponding references have been added to support sections 2.2&2.3. See manuscript page 3-5, Refs 29, 30, 31, 33, 34 in lines 116, 127, 133, 155 and 164.
- The format of Ref. 30. is not applicable to this article.
- FTIR was to confirm the successful synthesis of the four samples. Water absorption of the foamed SAP2 came from its porous structure confirmed by SEM with its increased the specific surface area. For SAP1 and SAP3, their surface cross-linking was mainly confirmed by the C-O-C asymmetric stretching vibration at 1170 cm-1 and the Al-O-C absorption peak at 576 cm-1, thus minimizing gel blockage and improving the performance.
- A new table has been added. There were only single modification techniques and few comparative studies of different modifications reported in existing literature. Majority of the latest researches focused on surface cross-linking, therefore, only the surface cross-linking reports have been compared in the table. See manuscript page 12, lines 368-380.
Thank you.
Sincerely yours,
Ting Xu
Graduate student of Shanghai University